# Exploring 3D Pelvis Orientation: A Cross-Sectional Study in Athletes Engaged in Activities with and without Impact Loading and Non-Athletes

**DOI:** 10.3390/jfmk9010019

**Published:** 2024-01-04

**Authors:** Georgios Glakousakis, Perikles Kalatzis, Dimitris Mandalidis

**Affiliations:** 1Sports Physical Therapy Laboratory, Department of Physical Education and Sports Science, School of Physical Education and Sports Science, National and Kapodistrian University of Athens, 17237 Athens, Greece; glakousakisg@phed.uoa.gr; 2Section of Informatics 1st Vocational Lyceum of Vari, Directorate of Secondary Education of East Attica, Hellenic Ministry of Education and Religious Affairs, 15122 Athens, Greece; periklesk@yahoo.com

**Keywords:** pelvis, standing position, female, athletes, sports, weight bearing

## Abstract

Female athletes subjected to various types of impact loading, especially over a long period of time, may experience changes in their pelvic orientation, which may affect their sport performance and increase the likelihood of injury. The aim of the present study was to determine whether female athletes involved in high-impact loading sports (HILS), odd-impact loading sports (OILS), and repetitive non-impact loading sports (NILS) demonstrate changes in pelvis orientation compared to non-athletes (NATH). Pelvic orientation was determined using Euler/Cardan angles, calculated from the coordinates of the right, and left anterior superior iliac spines and pubic symphysis via a novel method. Two-way ANOVA tests showed significant differences between groups for pelvis position in the frontal plane (*p* < 0.05), with HILS and OILS demonstrating greater pelvic obliquity compared to NILS athletes and NATH. Significant main effects were also obtained for directions within the sagittal plane (*p* < 0.001). Significant within-group differences were observed in sagittal pelvic position among female athletes engaged in NILS (*p* < 0.01) and non-athletes (NATH) (*p* < 0.05), with a greater anterior pelvic tilt compared to posterior. Our findings suggest that pelvis orientation in female athletes across sports is influenced by sport-specific impact loads, potentially affecting performance and injury occurrence.

## 1. Introduction

The pelvis, as part of the kinetic chain connecting the upper body and lower extremities, plays a key role in maintaining optimal body mechanics. The architecture and configuration of the bones comprising the pelvis, reinforced by strong ligaments and the multitude of muscles connected to it, contribute significantly to enhancing the overall mobility and stability of the body. This contribution is evident by (i) ensuring proper alignment and movement of adjacent joints, thus facilitating efficient muscle function [1,2], and (ii) absorbing and uniformly distributing the mechanical loads generated during movement activities in the surrounding soft tissue structures [3,4]. Ideally, this can be achieved when the pelvis is neutrally positioned. When viewed from the side, i.e., in the sagittal plane, the pelvis is considered in the “neutral” position when the anterior superior iliac spines (ASISs) are in the same vertical plane as the pubis symphysis (PS) [5,6]. This plane is also referred to as the anterior pelvic plane or triangle of Lewinnek [7]. In the frontal plane, the pelvis is neutrally positioned when the left and right ASIS are almost at the same level [8]. When the pelvis is viewed from above, i.e., in the transverse plane, it is considered neutrally positioned when the left and right ASIS are symmetrical and equidistant from the midline of the body [8].

In the general population, the neutral position of the pelvis often deviates from what is conventionally considered ideal. Herrington showed that the pelvis was tilted anteriorly by no more than 7° in most of the asymptomatic population studied [9]. In a recent study, Moharrami et al. identified pelvic obliquity ranging from 0° to 5.6° in male and female individuals considered part of the normal population [10]. In essence, the outcomes of these studies suggest that the pelvis in the general population undergoes positional shifts influenced by both internal forces from capsuloligamentous structures and muscles, as well as external impacts arising from daily activities such as interactions with the ground. The orientation of the pelvis, that is, the direction to which the pelvis is pointed, may be affected further by forces generated from various intrinsic factors such as musculoskeletal deviations in the spine (e.g., scoliosis) and lower limbs (e.g., leg length discrepancy), as well as pathologies and/or injuries (e.g., spine trauma) [11,12]. Moreover, the orientation of the pelvis can be influenced beyond what is typically observed in the general population when additional external loads are applied, thereby exacerbating their impact [13,14]. These loads can cover a spectrum of impact, ranging from excessively high—such as those occurring momentarily in falls or traffic accidents—to moderate or low, like those applied occasionally or repeatedly during sporting activities [8]. The impacts of these forces on pelvic orientation are insidious and often become noticeable when symptoms emerge after an impending injury.

Inevitably, any deviation in pelvic position, regardless of the plane in which it occurs, can potentially compromise sports performance, and increase the likelihood of injury [15,16,17,18,19,20,21,22]. Anterior pelvic tilt decreases both the upper and lower parts of rectus abdominis EMG activity while over-activating the rectus femoris, as opposed to the neutrally positioned or posteriorly tilted pelvis [15]. This muscular imbalance, combined with imbalances between other antagonistic muscle groups of the core (e.g., trunk and hip extensors), may compromise the natural lumbar lordotic curve, promoting an unbalanced distribution of forces both locally and globally and the development of musculoskeletal pain syndromes (e.g., low back and sacroiliac joint pain) [16,17,18,19]. Excessive anterior tilt has also been associated with an increased likelihood of hamstring [20], anterior cruciate ligament injuries [21], and patellofemoral pain [22]. This is probably not unjustified if we consider the resultant misalignment in the lower extremity joints that occurs with excessive anterior pelvic tilt, such as genu valgus/femoral internal rotation/ankle inversion [23].

In this context, several researchers have explored pelvic position in numerous prior studies aiming to discern the potential impacts of sports-induced loading on pelvic dynamics. Female gymnasts, who occasionally receive high-impact loading forces, have demonstrated less anterior tilt compared to non-athletes [24]. In contrast, male canoe/kayakers, who do not experience impact loads, and tennis players, who encounter impact loads due to the directional changes inherent in their sport, also called odd impact loading, exhibit no significant changes in pelvic tilt in the sagittal plane [25]. Other authors reported that athletes involved in high ground impact loading sports such as those experienced during maximal vertical jumps (e.g., volleyball) or odd impact loading (e.g., soccer, handball) can have varying effects on the frontal and transverse pelvic position in young male and female athletes, with results showing potential influences and no distinct effects [26,27,28,29,30,31]. Furthermore, unilateral or odd-impact applied loads in laterality-dependent sports such as field and ice hockey have been found to affect pelvic symmetry in female athletes [32]. This effect may manifest either by increasing pelvic obliquity or by iliac anterior/posterior rotation when compared with athletes in non-laterality-dependent sports and non-athletes [32]. Other investigators have demonstrated the impact of exercise-induced forces on pelvis position in young individuals who underwent exercises involving either unilateral or bilateral high-impact loads on the lower extremities, such as jumping down with landings on one or two feet [33].

While there is extensive research on the impact of loading patterns inherent in different sports and activities, the extent to which these sport-induced forces are influenced by the unique characteristics associated with each gender remains uncertain. The determination of pelvic position in female athletes requires focused attention due to the distinctive biomechanical challenges arising from the anatomical morphology and mechanical features of their pelvises [34,35]. Investigating this aspect can provide valuable insights into the prevention and management of gender-specific injuries in female athletes, recognizing the importance of tailoring interventions to the distinct biomechanical characteristics of the female pelvis within different sporting disciplines. Hence, the primary objective of the current study is to assess pelvis orientation in female athletes participating in sports characterized by varying degrees of sport-induced impact loading.

## 2. Materials and Methods

### 2.1. Participants

Pelvic orientation was investigated in sixty-five elite female athletes who systematically (>4 training sessions/matches per week) performed and trained for >8 years in impact and non-impact loading sport activities. Female athletes were selected based on the type of loading associated with the performance and training of the specific sport in which they were engaged, which was decided in advance, ultimately forming (i) a high-impact loading sports group (HILS), which consisted primarily of volleyball players (*n* = 25), (ii) an odd-impact loading sports group (OILS), which was comprised of soccer players (*n* = 22), and (iii) a non-impact loading sports group (NILS), which included swimmers and water-polo players (*n* = 18). High-impact loading sports were considered those requiring maximal vertical jumps and accompanying ground impacts; OILS involves turns and stops while spurting/running and accompanying ground impacts, while NILS required muscle forces occurring during long-lasting performances without ground impacts [36]. These impact loading types were chosen because of their diversity in direction, intensity, and frequency of occurrence, covering a wide spectrum of sports that can potentially affect pelvis orientation. In addition, they have also been suggested as three of the five almost distinct types of exercise loading that along with the high-magnitude and repetitive type of low-impact loading, are related, although not exclusively, to bone characteristics [36]. Twenty-six non-athletes of approximately the same age who did not regularly participate in sports were also included in the study to establish baseline measurements of pelvic position, to control for normal variability in pelvic position that may exist in the general population, and to determine whether the observed changes (if any) in females’ pelvic position can be attributed to sport-specific or normal loading.

The participants’ habitual physical activity level was determined using the Greek version [37] of the modified questionnaire developed by Baecke [38]. Both athletes and non-athletes participated in the study provided they had no previous lower extremity or spine injuries/operations and/or no excessive skeletal deviations such as scoliosis (<5° trunk rotation on the Adams test), leg length discrepancy (>0.5 cm), or overpronation/oversupination of the feet based on the Foot Posture Index-6. The study protocol was approved by the Institution’s Research Bioethics Committee (No. 1302/14-07-2021), and all participants gave written consent after being informed about the experimental procedure and the aim of the study.

### 2.2. Testing Procedure

Female athletes who were deemed eligible to participate in the study visited the sports physiotherapy facilities on a single occasion. An innovative and non-invasive method was used to determine pelvis orientation in three-dimensional space that accounted for possible inter-individual differences in the anatomy of the pelvis by calculating the Euler/Cardan angles based on the *x*-, *y*-, and *z*-axis coordinates of the right and left anterior superior iliac spines (ASISs) and the pubic symphysis (PS). Calculations of Euler/Cardan angles were performed using a specially developed software program written in Python. The software was designed to also allow the inclusion of the thickness of the overlying tissue of each bone landmark, as their coordinates were defined based on their projections on the body surface [39]. The location of the pelvis landmarks on the surface of the body and the determination of their coordinates were performed anthropometrically using a metric ruler and two commercially available digital laser distance meters (PLR 25 Digital Laser Measure, Bosch, Leinfelden-Echterdingen, Germany, Figure 1), which were placed in a device specially designed and constructed for the purpose of the study. The metric ruler was placed alongside the upper surface of a rectangular wooden box, which encased a vertical metal arrangement mounted on runners to enable its longitudinal displacement. The two laser distance meters were installed horizontally and vertically on an aluminum base fixed on an apparatus that was designed to slide on the vertical metal arrangement. The rectangular wooden box and a rotating platform were fixed to two opposite sides of a square-shaped wooden platform [39] (Figure 1).

Measurements were carried out with each subject (i) standing barefoot on the rotating platform in a relaxed upright posture while keeping her gaze on the horizon, (ii) the feet shoulder-width apart and parallel to each other, (iii) the heels touching a metal strip, and (iv) the index finger of the dominant upper limb touching a telescopic rod adapted to a stadiometer to reduce body sway [40]. Each bony pelvis landmark was first located with the laser beam of the horizontally placed distance meter by adjusting its position vertically and longitudinally via the sliding apparatus and vertical metal arrangement on which it was mounted, respectively. The coordinates on the *y*- and *z*-axis of each pelvis landmark were defined by measuring their distance tο the horizontally placed distance meter and their distance to the ground, as measured by the vertical distance meter, respectively. The coordinate on the *x*-axis was defined by marking the position of the vertical metal arrangement on the metric ruler that was placed alongside the upper surface of a rectangular wooden box using a built-in metric indicator (Figure 1).

Pelvis bony landmarks were detected using a 10LB linear probe connected to a LOGIQ 3 Basic diagnostic ultrasound (GE, Chicago, IL, USA) with each subject in a supine position. The examiner holding the ultrasound probe parallel and perpendicular to the spinal column for the detection of ASISs and PS, respectively [41,42] identified the bony prominences on the ultrasound screen and captured their optimal image by pressing the ultrasound probe against the skin with the least possible compression. A metal element (clip), which was placed between the ultrasound probe and the skin, was used to create interference with the ultrasound signal at the desired bony landmark and a cross-shaped impression where the interference was indicated by pressing the clip against the skin. The impression was finally marked with a paper sticker (Figure 2A–E). The thickness of the overlying tissues was measured twice at the points indicated by the interference just above the right and left ASIS as well as the right and left superior pubic rami, which were averaged for the PS. The mean of two measurements was considered in the analysis.

### 2.3. Validity and Reliability of the Testing Procedures

For consistency, both data related to the coordinates of the bony landmarks of the pelvis and the thickness of the overlying tissues were recorded by the first author (G.G.). Moreover, the reliability of the coordinate identification was assessed by repeating the procedure on two consecutive occasions within the same day in 11 athletes. The time between the two measurements was enough to return the skin to its original state after the impression was made on it during the ultrasonographic detection of the pelvis landmarks on the first occasion. The ICC (3,1) ranged from 0.978–0.999 and the SEM from 3.44–8.02 mm.

The intra-examiner and inter-examiner reliability of the procedure used to measure the thickness of the overlying tissues on the ASISs, and PS was also assessed by the first author (G.G.) and the third author (D.M.) of the manuscript in thirty athletes. The ICC (3,2) for intra-examiner reliability and the associated SEM ranged from 0.996–0.997 and 0.13–0.15 mm, respectively. The ICC (2,2) and SEM for inter-examiner reliability ranged from 0.960–0.984 and 0.35–0.45 mm, respectively.

The algorithm used to determine pelvis orientation was validated in a previous study by calculating the Euler/Cardan angles of predefined positions of an anatomical model in the sagittal, frontal, and transverse planes [39]. The ICC (3,1) was excellent (1.0), and the limits of agreement calculated between the predefined and the calculated positions were, in general, less than ±1.0°.

### 2.4. Statistical Analysis

The normality of the data distribution was examined with the Shapiro–Wilk test and by visually observing the Q–Q and box plot graphs. The Levens test for homogeneity was performed to assess the equality of variances among the groups.

Three 2 by 4 factorial ANOVA tests were undertaken to explore variations between pairs of directions within each plane (anterior–posterior tilt, left–right obliquity, and left–right rotation) as well as across groups (HILS, OILS, NILS, and NATH) concerning sagittal, frontal, and transverse pelvic positions. Within-group comparisons were conducted using post-hoc analysis with a Bonferroni adjustment. Pelvic orientation, as determined by all possible combinations of pelvic positions at all planes and directions within each plane between athlete and non-athlete groups, was analyzed using Fisher’s exact test. Fisher’s exact test was employed to compare categorical data due to the presence of expected counts in some cells that were less than five. Data analysis was performed using SPSS 28.0 (IBM Corp, Armonk, NY, USA), and the level of significance was set at α = 0.05.

## 3. Results

The demographic, anthropometric, and training characteristics of the athletes and non-athletes who participated in the study are presented in Table 1.

The two-way ANOVA yielded a non-significant main effect (reflecting between-group differences) for the participants’ groups (F = 1.12, *p* = 0.347, partial η^2^ = 0.039). However, a notable main effect emerged for the direction of pelvic tilt in the sagittal plane (F = 12.59, *p* < 0.001, partial η^2^ = 0.132), indicating that anterior tilt surpassed posterior tilt. The interaction between participants’ groups and pelvic tilt direction did not reach significance (F = 1.85, *p* = 0.144, partial η^2^ = 0.063). Post-hoc tests examining within-group differences revealed significantly greater anterior tilt compared to posterior tilt exclusively for the NATH group (10.2° ± 5.5° vs. 2.3° ± 2.8°; *p* < 0.05) and the NILS group (11.6° ± 5.9° vs. 2.7° ± 1.9°, *p* < 0.01). The anterior tilt compared to the posterior tilt for HILS athletes was 7.9° ± 7.1° vs. 6.2° ± 4.8°, and for OILS athletes, it was 10.4° ± 5.3° vs. 8.6° ± 7.0° (Figure 3).

Statistical analysis revealed significant between-group differences regarding pelvis obliquity (F = 3.436, *p* = 0.021, partial η^2^ = 0.110). Non-significant were the main effects for the direction of pelvic obliquity regardless of the group (F = 0.215, *p* = 0.644, partial η^2^ = 0.003) and the interaction between the groups of participants and the directions of pelvic obliquity (F = 0.542, *p* = 0.655, partial η^2^ = 0.019). Post-hoc tests indicated non-significant within-group differences concerning the direction of pelvic obliquity. Right-to-left pelvic obliquity was 1.5° ± 1.1° vs. 1.1° ± 1.0° for NATH, 1.0° ± 0.6° vs. 1.2° ± 1.0° for NILS athletes, 2.7° ± 2.0° vs. 2.0° ± 2.2° for HILS athletes, and 1.9° ± 1.2° vs. 2.1° ± 1.2° for OILS athletes. (Figure 4).

The findings indicated no significant main effects for pelvic rotation across groups (F = 0.96, *p* = 0.417, partial η^2^ = 0.033) and for pelvis directions regardless of the group (F = 0.420, *p* = 0.520, partial η^2^ = 0.005). Furthermore, the interaction between participants’ groups and pelvic rotation did not attain significance (F = 0.336, *p* = 0.799, partial η^2^ = 0.012). The right-to-left pelvis rotation demonstrated by NATH was 2.2° ± 1.9° vs. 2.3° ± 1.0°, by athletes engaged in NILS 1.7° ± 1.2° vs. 2.0° ± 1.2°, in those involved in HILS 2.3° ± 1.4° vs. 3.0° ± 1.5°, and in OILS 2.5° ± 2.2° vs. 2.3° ± 1.8° (Figure 5).

The contingency table (Table 2) illustrates the association between pelvis orientation (combinations of pelvis positions at all possible directions, i.e., anterior, or posterior pelvic tilt, right or left frontal pelvic obliquity, and right or left pelvic rotation) and group of participants (HILS, OILS, NILS, and NATH). Fisher’s exact test revealed a statistically significant association between the orientation of the pelvis and the group of participants (Fischer exact = 30.42, *p* = 0.034). The effect size, as measured by Cramér’s V, was moderate (V = 0.326). Specifically, pelvis orientation, represented by sequences S1–S4, was more prevalent in NATH (*n* = 22 or 24.2% of participants), followed by NILS athletes (*n* = 14 or 15.4%), HILS athletes (*n* = 13 or 16.5%), and OILS athletes (*n* = 11 or 12.1%). Pelvic orientation, as represented by sequences S5–S8, is nearly identical in athletes engaged in HILS and OILS (*n* = 21 or 23.1% of participants) in conjunction with sequences S1–S4 (*n* = 26 or 28.6%). However, this pattern is not as predominant in NILS athletes and NATH (*n* = 9 or 8.8%). The observed frequencies and percentages calculated based on the total number of participants are presented in Table 2.

## 4. Discussion

Concerning the sagittal pelvis position, a key component of pelvis orientation, our findings indicate that impact loading, whether high, odd, or absent, did not significantly change pelvic tilt both across the studied groups and in comparison, to non-athletes. Moreover, HILS and OILS demonstrated nearly equal anterior and posterior tilt, each of approximately the same magnitude. This contrasts with the higher proportion of female athletes engaged in NILS and NATH who exhibited a more pronounced anterior tilt than posterior tilt.

The method used in the present study to determine pelvis orientation in the athletic and non-athletic populations prevents, to a great extent, direct comparisons of our findings with observations from other studies. Our method allowed us to determine the tilt of the pelvis from its established reference neutral position. This contrasts with other published studies that utilized methods and instruments (e.g., trigonometric functions, inclinometers), which defined the pelvis position based on the inclination of the line connecting the ASIS and PSIS relative to the transverse plane [9,24,43]. Nonetheless, variations in the skeletal anatomy of the pelvis among individuals, leading to differences in the locations of its bony landmarks, impose limitations on the applicability of these methods. The range of variability observed, spanning from 0° to 23°, in the angle formed by the line connecting the ASIS and the ipsilateral PSIS concerning the transverse plane, while the pelvis is stabilized in an anatomically referenced position (neutral), has the potential to influence the accurate determination of pelvic tilt [44]. In this context, an individual may be falsely identified with anterior pelvic tilt when, in fact, the pelvis is posteriorly tilted. This information suggests that comparing the results of our study with those of other studies, regardless of whether they agree or not, should be interpreted with caution, as many of them were derived from measurements or methods that are questionable. Taking this into consideration, our observations on pelvic tilt align with Herrington’s [9] findings, which showed that most of the healthy female population studied exhibited anterior pelvic tilt (75%). This corresponds with the greater incidence of anterior tilt observed in NATH participants in the present study (84.6%). However, it is noteworthy that the average anterior pelvic tilt reported in his study was approximately 7°, slightly lower than the average of around 9° observed in the non-athletic population of our study.

Of equal importance were also the findings related to the similarities in magnitude and occurrence of anterior and posterior pelvic tilt presented by athletes involved in HILS and OILS, as opposed to NILS and NATH, suggesting a sport-dependent influence on the sagittal pelvis position. This implies that impact loading may have a distinct character specific to each sport, as reflected in the observed variations in pelvic tilt. Previous studies support, to some extent, these findings. Female gymnasts, who occasionally receive high-impact forces, especially in floor exercises, have demonstrated less anterior tilt compared to non-athletes. This position was determined by considering the position of the ASIS and PSIS relative to the transverse plane [24]. The observed difference in pelvic tilt may be partially attributed to the natural tendency of healthy females to exhibit a posterior tilt, given that the line of gravity typically falls behind the greater trochanter [45]. This inclination may be further reinforced in female athletes, particularly those involved in sports with jumping and cutting elements, as they often integrate strength exercises for core muscles, including the abdominal muscles [46], which are part of the force couple responsible for controlling posterior pelvic tilt.

While previous cross-sectional studies have suggested no direct relationship between abdominal muscle strength and pelvic position [47,48], targeted training initiated at an early age and sustained over two years has been shown to result in a conscious improvement in body posture [49], manifested with a straighter trunk position and a subsequent effect on pelvis position. The current study suggests that a comparable transformative impact on a specific group of female athletes could be a factor in heightening their awareness of body positioning and control. This heightened awareness may contribute to their ability to effortlessly sustain a neutral or even a posteriorly tilted pelvis, particularly in dynamic movements, thereby allowing them to maintain an optimal position for activities such as jumps and cuts. The adoption of a more posteriorly tilted pelvis could also be facilitated by the increased flexibility of the hip flexor muscles, which are antagonists to the trunk flexors. Athletes acquire this ability through the inclusion of hip flexor stretching exercises, both as stand-alone activities and as part of their warm-up routine, aiming to improve jumping performance [50] and reduce the risk of injuries among these athletes [51]. In addition to these factors, considering the potential impact of sport-induced loading on sagittal pelvis position, it is worth acknowledging that some individuals in HILS and OILS may have chosen their specific activities based on the presence of a posterior-tilted pelvis, a condition that is not uncommon in healthy females [45]. The posterior tilt of the pelvis reduces the stress placed on the muscles that control its movement, such as the hamstrings, which, combined with the greater flexibility they present, makes them less prone to injury [52]. Indeed, female athletes are two to four times less likely than males to develop a hamstring injury [53,54], who in turn present a higher anterior pelvic tilt angle in comparison to non-athletes [55]. This self-selection factor could contribute to the observed patterns of pelvic tilt within these groups.

Our findings regarding the significantly greater anterior compared to posterior pelvic tilt by most of the athletes involved in NILS were possibly attributed to postural adaptations induced by prolonged involvement in in-water sports such as hyperlordosis [56,57], a deviation that coincides with an increased anterior pelvis tilt. Such adaptations may result from the alternating movements of the upper extremities that force the spine into hyperextension [58] and ultimately the pelvis into excessive anterior tilt. The high forces exerted by the latissimus dorsi, erector spinae, and rectus femoris during swimming activity may also create a force couple acting on the pelvis, which, in combination with a less effective force couple generated by the trunk flexors and hip extensors, rotates the pelvis into a greater anterior tilt [59]. Similar muscular imbalances are anticipated in untrained, non-athletic individuals, where a predisposition to weakness and inhibition of tonic trunk flexors and hip extensors, namely the rectus abdominis and gluteus maximus, coinciding with the tendency for tightness or shortness in hip flexors (such as the rectus femoris) and trunk extensors (such as the lumbar erector spinae), may lead to the adoption of a forward-tilted pelvis [60].

Our findings reveal significantly greater pelvic obliquity among the groups tested, with HILS and OILS demonstrating greater pelvic obliquity compared to NILS and NATH. These findings diverge from previous studies, which reported non-significant pelvic obliquity in young female gymnasts aged 7–11 years with 1–5 years of training and experience, when compared to their untrained counterparts [26]. Separate studies indicated that the frontal plane pelvic position remained unchanged in female adolescent volleyball players aged 13–16 years [30] and in female handball players aged 12–15 years [27]. The differences noted across these studies are likely attributable to variations in the biological and, consequently, the training age of the players. In the current study, participants were elite athletes who were older and had accumulated more years of training compared to the junior participants in the earlier studies. Even though some researchers demonstrated that 2 years of practicing volleyball in 13–14-year-old female players did not alter pelvis obliquity [31], there is evidence that alterations may eventually occur as young individuals engage in exercises involving either unilateral or bilateral high-impact loads on the lower extremities, including activities such as jumping down and landing on one or two feet [33]. Gnat et al. [33] noted in the aforementioned study that the most significant potential for inducing pelvic asymmetry, a term commonly used to describe the uneven alignment of the pelvic in both the frontal plane (lateral pelvic tilt) and the sagittal plane (iliac anterior/posterior rotation asymmetry) [19,61], occurred in instances of asymmetrical loads that were applied in the form of what is referred to as a “mechanical shock”—that is, a force characterized by a substantial impulse. Volleyball and handball athletes regularly experience such loads during landings, as a significant portion of their landings following a jump occurs on a single leg [62,63]. Continued exposure to these forces over an extended period may eventually influence the frontal alignment of the pelvis. For instance, the outcome of asymmetric loading resulting from lateral dominance, coupled with the requisite flexed and rotated trunk posture in unilateral OILS such as field hockey, ice hockey, or speed skating, tends to induce greater pelvic asymmetry in athletes participating in these sports compared to those in non-laterality-dependent sports and individuals who are not engaged in athletic activities [32].

Pelvic rotation in the transverse plane is a facet of pelvic orientation that has received limited investigation, resulting in a scarcity of data in the existing literature. A potential explanation for the absence of changes in pelvic rotation in the transverse plane following prolonged involvement in sports could be the lack of systematic monitoring or assessment during training and sports participation. Without regular and attentive evaluation, alterations in pelvic rotation may go unnoticed or undocumented. The methodology employed in our study allowed us to identify a subtle, although not statistically significant, increase in pelvic rotation among HILS athletes compared to OILS, NILS athletes, and NATH. Nevertheless, there were no significant bilateral differences observed in pelvic rotation. These findings are consistent with the results reported in earlier studies conducted with young female gymnasts, handball players, and volleyball players, revealing non-significant differences in pelvic rotation across most of the age groups tested [26,27,30]. Sports activities and exercises that do not substantially challenge or require changes in pelvic rotation in the transverse plane, along with biomechanical adaptations seen in sports emphasizing linear or primarily sagittal plane movements, may collectively contribute to the prevention of significant changes in pelvis rotation. It appears that participants in our study, engaged in sports such as volleyball, soccer, swimming, and water polo, were involved in activities that did not necessitate significant pelvis rotation, at least to the extent seen in other sports such as golf [64], martial arts [65], or baseball [66], as well as activities such as dancing [67].

### Study Limitations

Caution is warranted when generalizing our findings to other populations, considering the limitations associated with participant gender, the specific type of athletic activity involved in the sports under investigation, and the active years of engagement. The distinctive anatomical characteristics of the female athlete, including the shape of the pelvis [34], the mobility of adjacent joints, and the tension of pelvic ligamentous structures [35], limit the present findings to this specific gender. Differences in the impact of forces generated by sports, such as high-magnitude loading referring to maximally applied muscle forces in slow, well-coordinated movements without ground impacts (e.g., powerlifting) and repetitive low-impact loading referring to ground impacts during long-lasting running performances at a relatively constant speed, may result in changes in pelvic position that were not observed in the current study [36]. Extended periods of involvement in the performance and training of a specific sport, in comparison to the duration examined in the present study, may also contribute to noticeable shifts in body posture, including pelvis position [68].

Our findings are also limited to female athletes who do not exhibit excessive skeletal deviations in the sagittal or frontal plane that may potentially impact pelvic posture. Conditions such as idiopathic scoliosis have been associated with alterations in pelvis position in the sagittal [69], frontal [12,69], and transverse planes [70]. Similarly, leg length discrepancies of anatomical origin can contribute to pelvic asymmetry [11], and foot overpronation has been linked to an increase in anterior pelvic tilt [23]. Furthermore, it should be considered that the present measurements were performed without considering any pelvic deformities, which could affect pelvic symmetry, such as unilateral iliac hypoplasia [71].

## 5. Conclusions

The study’s findings reveal that pelvis orientation is influenced by impact loads in female athletes participating in diverse sports activities. Specifically, female athletes engaged in HILS and OILS demonstrated anterior pelvic tilt of the same magnitude and occurrence as posterior tilt. Athletes involved in NILS demonstrate a greater anterior pelvic tilt compared to posterior pelvic tilt, similar to the observed pattern seen in individuals categorized as NATH. Moreover, pelvis position in the frontal plane was different between female athletes and non-athletes, with those exposed to both high and odd impact loading demonstrating a more pronounced frontal pelvic position compared to NILS athletes and NATH. Exercise-induced pelvic deviations possess the potential to influence athletes’ performance and, over time, increase the risk of injury. In this light, coaches and clinical therapists could modify their training regimens by incorporating exercises that enhance the absorption of exercise loads, potentially mitigating the impact of pelvic misalignments, particularly for athletes presenting with frontal pelvic deviations. They could also implement compensatory measures to counterbalance sagittal plane deviations, especially in NILS athletes, potentially improving sport performance and reducing the likelihood of long-term injuries.

### Future Studies

Given the existence of a valid methodology that can assess pelvic position non-invasively with ease and reliability, researchers from all perspectives could focus in future studies on (i) monitoring changes in pelvic alignment over an extended period, considering the cumulative impact of sports participation and load exposure; (ii) examining the effect of impact loads on the pelvis across a broader spectrum of sports or athletic activities; (iii) establishing correlations between pelvic disorientation and the occurrence of musculoskeletal injuries; (iv) evaluating the effectiveness of targeted interventions, such as specific training programs or corrective exercises; (v) exploring the influence of age and duration of training on the impact loads experienced by the pelvis; (vi) investigating potential gender differences in pelvic responses to impact loads; and (vii) assessing functional outcomes associated with pelvic deviations, encompassing their impact on athletic performance, biomechanics, and overall health.

## Figures and Tables

**Figure 1 jfmk-09-00019-f001:**
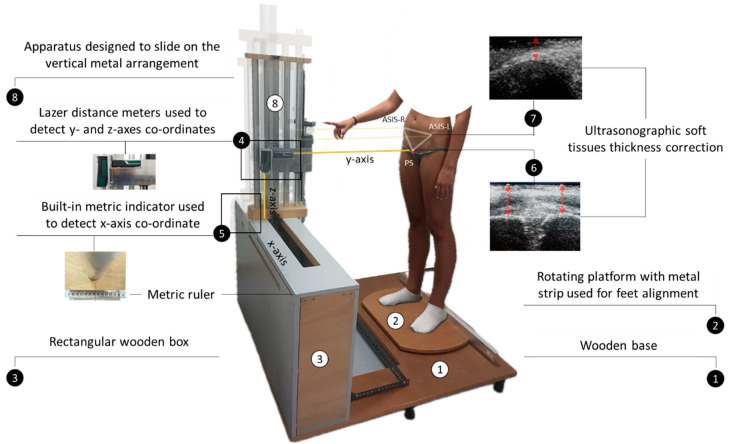
Device and procedure used to determine the coordinates of the pelvis bony landmarks.

**Figure 2 jfmk-09-00019-f002:**
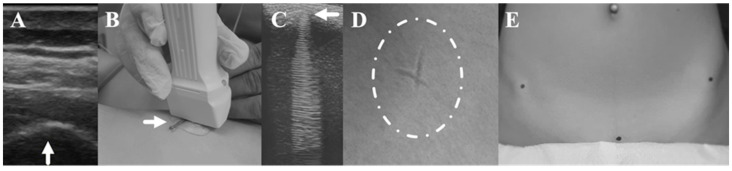
(**A**–**E**). Procedure for identifying pelvic bony landmarks. Ultrasound detection of the desired pelvic bone landmark (as shown by the arrow for ASIS) (**A**), metal element (shown by the arrow) placed between the ultrasound probe and the skin (**B**), interference created by the metal element at the level of the pubic symphysis (shown by the arrow), as depicted on the ultrasound (**C**), skin impression created at the interference site (**D**), impression marking with paper stickers (**Ε**).

**Figure 3 jfmk-09-00019-f003:**
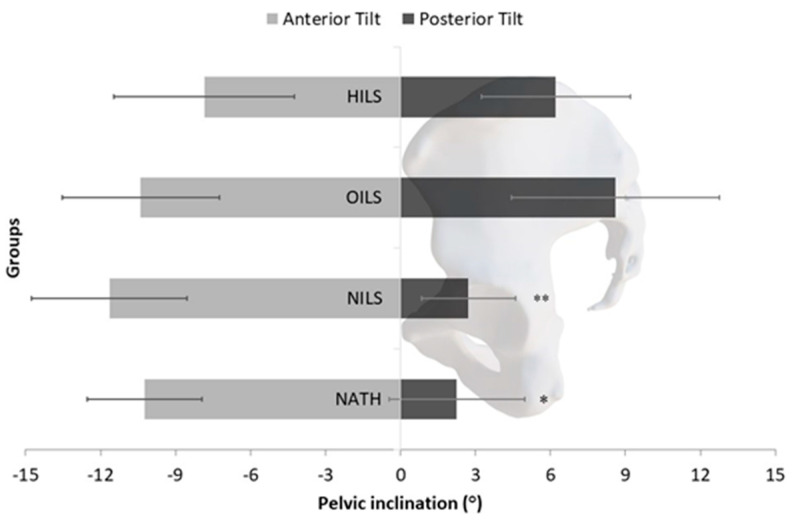
Mean values and 95% confidence limits of the anterior and posterior pelvic tilt for the female athletes involved in high-impact loading sports (HILS), odd-impact loading sports (OILS), and non-impact loading sports (NILS), as well as non-athletes (NATH). * *p* < 0.05: significant difference compared to anterior tilt; ** *p* < 0.01: significant difference compared to anterior tilt.

**Figure 4 jfmk-09-00019-f004:**
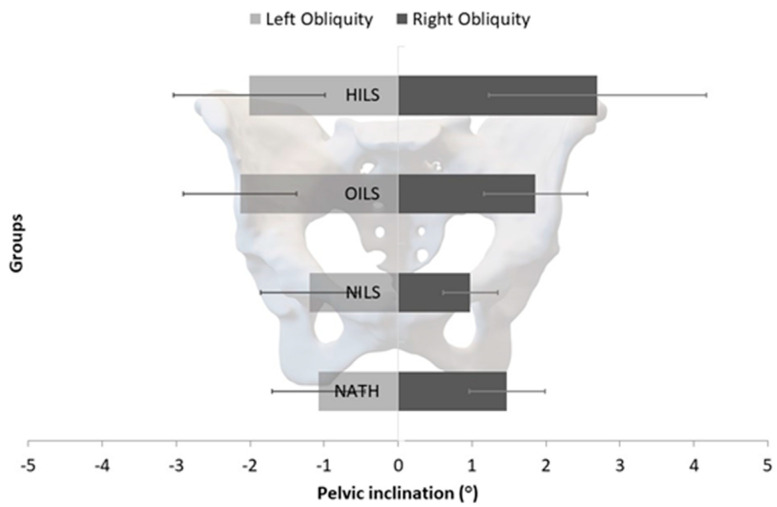
Mean values and 95% confidence limits of the left and right pelvic obliquity for the female athletes involved in high-impact loading sports (HILS), odd-impact loading sports (OILS), and non-impact sports (NILS), as well as non-athletes (NATH).

**Figure 5 jfmk-09-00019-f005:**
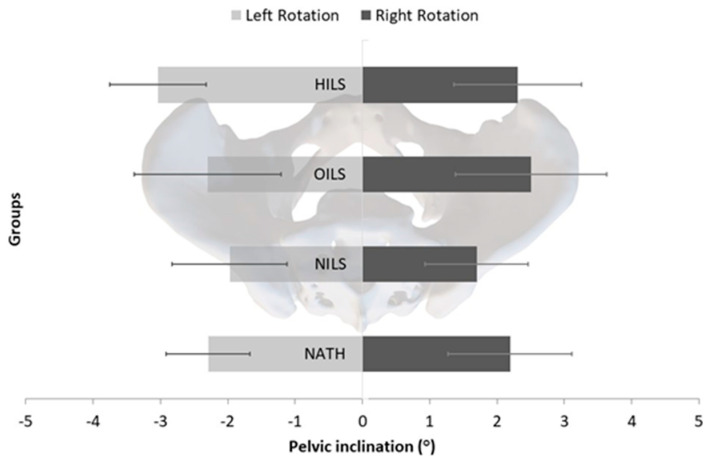
Mean values and 95% confidence limits of the left and right pelvic rotation for the female athletes involved in high-impact sports (HILS), odd-impact sports (OILS), and non-impact loading sports (NILS), as well as non-athletes (NATH).

**Table 1 jfmk-09-00019-t001:** Mean ± standard deviations of demographic, anthropometric and training characteristics of female athletes and non-athletes participated in the study.

	HILS (*n* = 25)	OILS (*n* = 22)	NILS (*n* = 18)	NATH (*n* = 26)
Age (yrs.)	24.6 ± 4.0	24.0 ± 4.0	25.4 ± 4.0	27.8 ± 3.6 *
Height (m)	1.8 ± 0.1	1.7 ± 0.1 **	1.7 ± 0.1 **	1.7 ± 0.1 **
Body mass (kg)	68.4 ± 8.2	63.2 ± 8.9	66.2 ± 9.8	59.6 ± 7.1 ***
BMI (kg·m^−2^)	21.6 ± 1.6	23.1 ± 3.0	22.7 ± 2.3	21.4 ± 2.6
HPAL (points)	10.2 ± 1.5	9.0 ± 0.2 ^††^	10.5 ± 1.5	7.7 ± 0.6 ^†^
Years active	12.9 ± 2.4	10.9 ± 2.5 ^†††^	13.3 ± 1.6	-
Age/years active	1.9 ± 0.2	2.3 ± 0.4 ^†††^	1.9 ± 0.3	-

Note: BMI = body mass index; HPAL = habitual physical activity level; HILS = high-impact loading sports, OILS = odd-impact loading sports, NILS = non-impact loading sports, NATH = non-athletes; * significant difference (SD) compared to HILS, OILS (*p* < 0.05); ** SD compared to HILS (*p* < 0.01); *** SD compared to HILS (*p* < 0.01); ^†^ SD compared to HILS, OILS, and NILS (*p* < 0.01); ^††^ SD compared to HILS, NILS (*p* < 0.05); ^†††^ SD compared to HILS and NILS (*p* < 0.01).

**Table 2 jfmk-09-00019-t002:** Occurrences of pelvic positions in absolute numbers and percentages of participants (in parentheses) across all possible combinations within each plane and direction for female athletes and non-athlete groups.

Sequence	Pelvis Orientation	Groups				
		HILS	OILS	NILS	NATH	Total
S1	APT–RPO–RPR	0 (0.0)	5 (5.5)	5 (5.5)	9 (9.9)	19 (20.9)
S2	APT–RPO–LPR	3 (3.3)	3 (3.3)	5 (5.5)	5 (5.5)	16 (17.6)
S3	APT–LPO–RPR	5 (5.5)	3 (3.3)	3 (3.3)	6 (6.6)	17 (18.7)
S4	APT–LPO–LPR	7 (7.7)	0 (0.0)	1 (1.1)	2 (2.2)	10 (11.0)
S5	PPT–RPO–RPR	1 (1.1)	1 (1.1)	0 (0.0)	1 (1.1)	3 (3.3)
S6	PPT–RPO–LPR	3 (3.3)	3 (3.3)	0 (0.0)	2 (2.2)	8 (8.8)
S7	PPT–LPO–RPR	3 (3.3)	3 (3.3)	2 (2.2)	1 (1.1)	9 (9.9)
S8	PPT–LPO–LPR	3 (3.3)	4 (4.4)	2 (2.2)	0 (0.0)	9 (9.9)
	Total	25 (27.5)	22 (24.2)	18 (19.8)	26 (28.6)	91 (100.0%)

Note: HILS = high-impact loading sports, OILS = odd-impact loading sports, NILS = non-impact loading sports, NATH = non-athletes, APT = anterior pelvic tilt, PPT = posterior pelvic tilt, RPO = right pelvic obliquity, LPO = left pelvis obliquity, RPR = right pelvic rotation, LPR = left pelvic rotation.

## Data Availability

The data presented in this study are available upon request from the corresponding author.

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
