# Peer review of "Exploring 3D Pelvis Orientation: A Cross-Sectional Study in Athletes Engaged in Activities with and without Impact Loading and Non-Athletes"

_jfmk, 2024, doi:10.3390/jfmk9010019_

Round 1

Reviewer 1 Report

Comments and Suggestions for Authors

Reviewer 2 Report

Comments and Suggestions for Authors

Introduction

These can be either momentary but high-intensity forces such as those that occur in falls or traffic accidents, or repeated moderate-intensity forces like those that generated when performing sporting activities [9].

It should be emphasized that the paper deals exclusively with repeated moderate-intensity forces like those that generated when performing sporting activities. Sports with high-intensity forces are for example Rugby, sports with falls are for axample Judo.

Coment 1

You analyze the forces that are transmitted to the surface, what about the forces that are generated in contact with the opponent (they are much larger and not rare)?

Sample

Sixty-five female athletes who systematically (>4 training sessions/matches per week) involved for >8 years in high-impact sports (HIS, n=25), odd-impact sports (OIS, n=22) and (iii) non-impact sports (NIS, n=18) as well as healthy female non-athletes (HNA, n=26) were included in the study. The HIS group included athletes whose sport required jumps with or without initial velocity such as volleyball, the OIS group included athletes whose sport required changes of direction such as soccer, while the NIS group included athletes whose sports did not induce ground reaction forces like swimming and water polo [31].

Coment 2

Terminology is a bit confusing, when you say high-impact sports we first think of martial arts or Rugby where there are brutal blows. Volleyball belongs to non-contact sports, the net divides the two teams and the opponents do not come into contact. That group should be called: Group of sports dominated by jumps. The odd-impact sports group refers to sports similar to football. Soccer, unlike volleyball, has contact in the game, which is why there are more injuries and it is classified as an odd-impact sport. That group should be called: Group of sports dominated by changes of direction. Water polo is a contact game with often brutal hitting and wrestling in the pool, it is in the group of non-impact sports. A group of sports where it is not possible to transfer the force of the legs to the surface.

It would be interesting to conduct such research on sports where large forces are transferred to the surface, such as weightlifting, judo or wrestling.

Coment 3

Specify which sports are included in which group, sports like volleyball or football are not enough. For example, when you say sports like volleyball, the reader thinks of football, both sports belong to collective sports games with a ball.

Coment 4

Water polo, volleyball and football are from the same group of sports, collective sports games with a ball. Considering that one group of sports is involved, it is not right to generalize "Long-Term Sports-Induced" in the title.

Discusion

Another finding that also contradicts ours based on which female gymnasts demonstrated less anterior tilt compared to non-athletes, may also be misleading, as the authors in this study determined the position of the pelvis trigonometrically considering the relative position of the aforementioned bony landmarks [17].

Coment 5

Gymnastics does not belong to the same group of sports, this may be due to a different load structure or due to the influence of flexibility, which is extreme in gymnastics.

Final coment

The problem is that the sample does not come from different sports but from one group of sports. This is not visible in the title, introduction and discussion. Changes should be made wherever it is generalized that it is the influence of sports on the anatomy of women, it is the influence of a certain group of sports.

Round 2

Reviewer 2 Report

Comments and Suggestions for Authors

The paper has been significantly improved, and can be published.